# Sustainability through Humility: The Impact of Humble Leadership on Work–Family Facilitation in the U.S. and Japan

**Soyeon Kim [1], Neena Gopalan [2],\* and Nicholas Beutell [3]**

[1] Faculty of International Social Sciences, Gakushuin University, Tokyo 171-0031, Japan; soyeon.kim@gakushuin.ac.jp
[2] School of Business, University of Redlands, Redlands, CA 92373, USA
[3] LaPenta School of Business, Iona University, New Rochelle, NY 10801, USA; nbeutell@iona.edu
\* Correspondence: neena_gopalan@redlands.edu

**Abstract:** This study examines the influence of leader humility on work–family facilitation (WFF) in the U.S. and Japan by exploring the mediating roles of the four dimensions of psychological empowerment (meaningful work, autonomy, competency, and impact) on this relationship. Drawing from a sample of 392 Japanese employees and 132 U.S. employees, our findings suggest that leader humility is positively related to WFF in both cultural contexts. Meaningful work and departmental impact emerge as significant mediators in both cultures, while the mediation effects of autonomy and competency are valid in Japan only. An additional test reveals that meaningful work is the most significant mediator in both countries, underscoring the pivotal role of leader humility and meaningful work in enhancing WFF. The study adds to the growing literature on the beneficial effects of leader humility on sustainable organizations, while offering insights into improving employee wellbeing and work–life interactions across diverse cultural contexts.

**Keywords:** leader humility; sustainable organizations; psychological empowerment; meaningful work; work–family facilitation; cross-cultural comparisons





## 1. Introduction

The purpose of this study is to test the role of leader humility on outcomes of both employee and organizational interests in two different cultures, the U.S. and Japan. Specifically, we empirically test the model of the effect of leader humility on work–family facilitation (hereafter, WFF), mediated by the dimensions of psychological empowerment. Using time-lagged data collection from both USA and Japan, the study primarily tested the hypothesis that leader humility can assist in employees experiencing heightened psychological empowerment which also leads them to have better WFF.

We define leader humility as "a willingness to view oneself accurately, an appreciation of others' strengths and contributions, and teachability, or openness to new ideas and feedback" [1]. The term leadership conjures qualities such as courage, taking initiative, power, and other related terms. However, recently, the virtue of 'humility' has been recognized as vital for leadership, signifying the prevalence of traits that transcend self-interest [2]. This trait is reflected in the leader's willingness to take a more realistic view of themselves while being open to others' feedback as well as having an appreciation of others' strengths and ideas [1]. Humility allows leaders to view themselves more realistically including being aware of their limits and shortcomings, willingness to solicit feedback as well as acknowledging and appreciating the values and strengths of others [1]. Several studies allude to the potential encouraging effects of leader humility. For example, a study on a Chinese sample found that servant leader humility affected follower engagement [3], while a Pakistani study noted that leader humility can have an inspiring impact on employee's success in their projects [4]. Further studies, e.g., [5,6] have also reported the positive effect of leader humility in the workplace.

Importantly, leader humility is related to organizational sustainability in a number of ways. Humble leaders acknowledge their own limitations and shortcomings by creating an environment with features such as shared learning, collaboration, and open communication that are essential aspects of sustainability [7]. Humble leaders also acknowledge the inputs and perspectives of others as well as give credit to others [1]. Such behaviors increase work engagement, task meaningfulness, and autonomy by increasing employee satisfaction and reducing turnover, thus creating a more productive and sustainable workforce. Humble leaders also focus on long-term goals related to sustainability, create a safe environment where teams can flourish, encourage innovation, and embrace ethical behavior and corporate social responsibility. Leaders also understand that employees have lives outside of work and thereby support the whole person [7], one of the keystones of sustainability.

This study has three major goals. First, despite the potentially positive impacts of leader humility in organizations, its influence on individual employees and the psychological process through which it leads to employees' work outcomes have been rarely studied. The present study notes the significance of humility and the lack of studies that have addressed this specific trait of leaders. Hence our study is innovative in the sense that it explores the complex dynamics of leader humility on WFF and other important outcomes. The present study notes the significance of humility and the lack of studies that have addressed this specific trait of leaders. We intend to clarify the impact of leader humility on synergies of employees' work and life participation including its psychological processes. This cross-cultural study also recognizes the importance of leader humility in sustainable organizations and sustainable employee wellbeing. Building on Social Information Processing theory (hereafter, referred to as SIP) [8], the study suggests that the positive effect of leader humility on WFF and the mediating effect of employees' psychological empowerment on the relationship between leader humility and WFF.

Second, the concept of leader humility has not been well received, rather it was reported that it has a negative connotation in some cultural contexts in which self-expression and argument are valued. Additionally, as some variables such as gender and age confound the relationship between leader humility and employees' behavioral outcome variables [9,10], the true effect of leader humility on employees is still controversial. However, as work force diversity and organizational inclusiveness are critical matters to achieve sustainable management, by attracting and retaining talents, humility is gaining attention in leadership studies in the Western context as well and is considered an important aspect of effective leadership [11,12]. Therefore, the topic of leader humility and its relationship to outcomes can help to confirm its practical applicability and significance in organizations that are faced with changing demographics and organizational norms.

Third, most studies on leader humility have been conducted in an Eastern context, e.g., ref. [13] with very few studies conducted in other parts of the world. Addressing this empirical gap will broaden our understanding in how leader humility is interpreted and accepted to employees and also how it influences employees depending on the cultural context. Thus, the comparative study will provide insights into the impact of leader humility and its presumed psychological processes across cultures. Although the humility concept is embraced more fully in the Eastern context, this trait is increasingly expected of leaders in the West as well because of the social movement toward valuing diversity and inclusiveness [14]. Cross-cultural studies on this topic are encouraged and the current study is one such study, where we attempt to explain the effects of leader humility in Japan and the United States and its impact on outcome variables in both cultural contexts.

*Theoretical Background*

We base this study on SIP theory [15] which underlines the crucial role of leaders serving as a source that gives cues about organizations to their employees. Employees subsequently interpret the information and shape their attitudes and behaviors toward organizations [15]. SIP theory underlines the importance of context and outcomes of one's previous decisions, in contrast to rational decision-making. It focuses on how individuals

form attitudes in the workplace social context. Leaders tend to be significant sources of information in the workplace [16] and tend to influence their employees' attitudes and actions. It is highly therefore probable that leader humility can 'transfer' to employees and bring a similar sense of humility in them [17]. SIP has often been used in research work explaining leader humility and its impact on employees, e.g., [8].

## 2. Literature Review and Hypothesis Development

*Leader Humility and Its Effect on WFF*

Below, we first provide a quick overview of leader humility, followed by a literature review on WFF and the mediator variable of psychological empowerment.

*Leader humility*—There is a dearth of empirical research on leader humility, particularly in terms of its impact on the effectiveness of employees and organizations [10]. Some studies allude to humility as a weakness for leaders [18]. Also, some found its contingent effects on followers' demographic characteristics such as age and gender [9,10]. That is, studies have found that the effect of leader humility on employees may differ depending on employees' gender and their age similarity with their leader [9,19]. However, studies have, in general, agreed on the potential encouraging impact of leader humility on organizational members and its managerial importance [4,8,10,13] as well as on positively impacting employee attitudes and behaviors. Leader humility serves as an important source of organizational information and facilitates similar adaptation mechanisms, fostering employee humility, and thus enhancing employees' wellbeing at work [8]. Humble leaders are appreciative of others and open to others' feedback [4,8,10,12]. The nature of leaders can potentially build trust among employees and enable them to feel psychologically satisfied with their work, elevating their work engagement and chances of being successful in their work [8]. The positivity and satisfaction about their work help in the family domain as well (WFF), explained below.

*WFF*—As a positive spillover effect of work on the family, WFF occurs "when involvement in work results in a positive emotional state or attitude which helps the individual to be a better family member" [20] (p. 140). WFF is sometimes referred to as work–family enrichment [21], positive spillover [22], or work family synergy [23] and is the process by which participation in one role (work or family) enriches or enhances the quality of life in the other role (family or work) [21]. This facilitation can occur in two directions: work-to-family (W → F) and family-to-work (F → W) [24]. Work-to-family facilitation refers to the extent to which experiences at work improve the quality of family life. For instance, skills, behaviors, or positive moods gained from the workplace may enhance an individual's family interactions or relationships [25]. Conversely, family-to-work facilitation occurs when positive experiences in the family role improve the quality of work life. For example, the support or happiness derived from family life might lead to increased job satisfaction or improved performance at work [26].

Many factors can influence WFF, including organizational culture, job characteristics, and personal resources [27]. For instance, workplace policies supportive of work–life balance (like flexible work arrangements), job autonomy, and high-quality relationships with coworkers or supervisors can facilitate WFF [21,24]. Comprehending and enhancing WFF is critical as it is linked to numerous positive outcomes, such as increased job and life satisfaction, improved job performance, better physical and mental health, and reduced turnover intentions [25,26].

*Leader humility and WFF (WFF)*: Humble leaders are known for their openness to feedback, readiness to appreciate others' contributions, and willingness to admit mistakes and foster an environment that can assist in helping employees manage their work and life activities, thereby facilitating work–family integration [10]. Humble leaders typically engage in fair treatment of employees, communicate clearly, and share decision-making, which may lower work stress levels and further promote a balanced work–family dynamic [21]. Humble leadership is shown to enhance employee satisfaction and performance, and these positive work experiences can spill over into a more harmonious home life, further en-

riching the work–family relationship [10]. A humble leader exhibits transparency in his or her demeanor and openly shows appreciation for their employee's contributions [12], thus benefitting employee's overall wellbeing [8]. Such a leadership style tends to improve employee's personal qualities such as adaptation and resilience [28]. This in turn, we argue, can also lead to better adaptation and fulfillment of responsibilities in one's work and family domains.

We hypothesize that the relationship between humble leadership and WFF may exist in both the U.S. and Japan, despite their cultural differences and desired leader prototype. In the U.S., individualism is a core value and business environments tend to be rather competitive and performance oriented. This supposes that leaders whose traits are charismatic and masculine are more desired and perceived to be effective. However, the societal changes in the United States toward advocating and appreciating diversity and inclusiveness [14] may call for humbleness and openness among leaders. A 2018 study [29] noted that humble CEOs had lower turnover rates among their employees and overall better company performance. Research on leaders' humility on U.S. leaders and its impact are still in its infancy, which necessitates more empirical work in this domain. We expect that humble leadership might enhance personal satisfaction and positive feeling about work, subsequently improving work–life facilitation in the U.S.

Meanwhile, the Japanese culture emphasizes respect, harmony, and group orientation [30,31]. Leadership behaviors are catered more toward relationships that display individual care and concern and encourage employees' participation [32]. With high moral standards, leaders are expected to be humble by valuing others and their opinions [31]. All these highlight the fact that humble leaders value other-focused approaches rather than self-focused ones. Such leaders appreciate those behaviors congruent with the prototypic traits that Japanese people expect from desired leaders. Therefore, such equivalence between the leader prototype and the behaviors of humble leaders will enhance the relationship quality between leader and subordinates, and employees' trust in leaders, leading to positive work outcomes of job satisfaction and work motivation [33]. Such positive experiences and work motivation at work tend to have a positive facilitation effect on the family. Building on the rationale described above and tenets of SIP theory, we hypothesize that leader humility positively influences WFF both in the U.S. and Japan. That is, when employees observe their leaders engaging in humble behavior, they are also more likely to incorporate such behaviors into their own. Thus,

**H1.** *Leader humility is positively related to WFF both in the U.S. and Japan.*

*Psychological Empowerment*: Psychological empowerment of employees refers to cognitively increased motivational energy toward work that is manifested in four dimensions: meaning, competence, self-determination, and impact [34]. Work meaningfulness is defined as the value of work goals and purpose. Competence is an individual's belief in their work ability and skill to complete work assignments. Self-determination refers to an individual's sense of autonomy in choosing and regulating their work processes and actions. Impact indicates an individual's belief that he or she can influence the strategic, administrative, and operational outcomes of organizations [35]. This elevated work motivation in the four dimensions of psychological empowerment can result in employees being more passionate toward their work, contributing to increased work engagement and organizational commitment [36,37].

Leader humility may change employees' psychological approaches toward their work. Employees feel that their work is valued, which contributes to a feeling of strengthened psychological empowerment. Psychological empowerment is influenced by the organizational environment and by interactions with the leaders [36,38]. Leaders give cues about an organization's policies and direction on treating employees, and such information is processed by employees, shaping their work attitudes [15].

Leader humility serves employees to have a broader and more holistic view of the work, specifically in a project context [39]. Research has shown a positive relationship

between a leader's humility and the psychological empowerment of his/her employees [40]. Humble leaders' open approach and attitudes encourage employees to share their job-related information and collaborate with each other [2]. Such behaviors let employees perceive the shared vision of the organization and the way that such vision can be achieved. Consequently, they gain a clearer view of their work and comprehend the meaning and significance of their work. A leader's power-sharing habit may inspire employees and those who feel their contributions are valued tend to be better performers [41]. Thus, leader humility may psychologically empower employees.

We hypothesize that such elevated meaning at work may have synergistic effects on managing family. The literature suggests that leader humility can enhance work meaningfulness, which in turn increases WFF. Humble leaders foster a positive work environment [10] that can increase work meaningfulness [6]. Leader humility can enable employees to have a more positive and holistic view of one' work [39]. Perceiving work as meaningful can enhance job satisfaction and reduce stress, thereby positively impacting WFF [21]. While cultural nuances may affect these relationships, the core assumption is that leader humility can contribute indirectly to WFF by enhancing the sense of meaningfulness in work which is often elevated in a positive work environment and can augment WFF [6]. Providing work resources including meaningful work can facilitate a transfer of positive emotions and attitudes from work to family [42], thereby enhancing WFF [43]. This suggests that humble leadership can indirectly enhance WFF by amplifying the work's significance. SIP theory stipulates the influence of leadership on employee's actions and perceptions. Here, a leader's humble behaviors can indirectly benefit WFF by increasing the sense of work significance. This leads to our Hypothesis 2:

**H2.** *Meaningful work mediates the relationship between leader humility and WFF in the U.S. and Japan.*

Next, we posit that work competency mediates the relationship between leader humility and WFF. Humble leaders often foster environments that encourage learning and skill development [10], thus potentially enhancing employees' work competency. Such work dedication can lead to work effectiveness [44] through intrinsic motivation at work. Humble leaders also tend to value the contributions of their employees and are more likely to delegate authority [2], both of which boost employees' confidence [45] and competence. As per the SIP theory, employees tend to assimilate better attitudes and behavior through their leader's humble actions. Psychological competence, enabled by the leader's actions, can also have a positive bearing on one's overall psychological wellbeing and happiness [46]. Increased competency can lead to higher job satisfaction, better performance, and reduced stress [47], which can positively affect WFF [21]. Based on the above, we propose that:

**H3.** *Work competency mediates the relationship between leader humility and WFF in the U.S. and Japan.*

Similarly, we propose that work autonomy mediates the relationship between leader humility and WFF. Humble leaders often foster environments that empower employees [10], thereby enhancing their work autonomy. Such employees tend to be more confident [48] and tend to put forth increased efforts in accomplishing their work tasks [49] and demonstrate efficient performance [50]. The SIP theory stipulates the role of leaders in bringing in a newer attitude and behavior. We contend that such positive experiences at work can have an effect not only at work but also in one's family domain as well, bringing in WFF [21].

**H4.** *Autonomy mediates the relationship between leader humility and WFF in the U.S. and Japan.*

Finally, we argue that employees' departmental impact mediates the relationship between leader humility and WFF. Humble leaders often foster environments that facilitate collaborative efforts and collective success [10], thereby enhancing the departmental impact. Greater departmental impact can lead to higher job satisfaction and a sense of

accomplishment [51], which may positively affect WFF [21] since leader humility can indirectly contribute to WFF by amplifying employees' departmental impact.

**H5.** *Departmental impact mediates the relationship between leader humility and WFF in the U.S. and Japan. The research model is shown in Figure 1.*

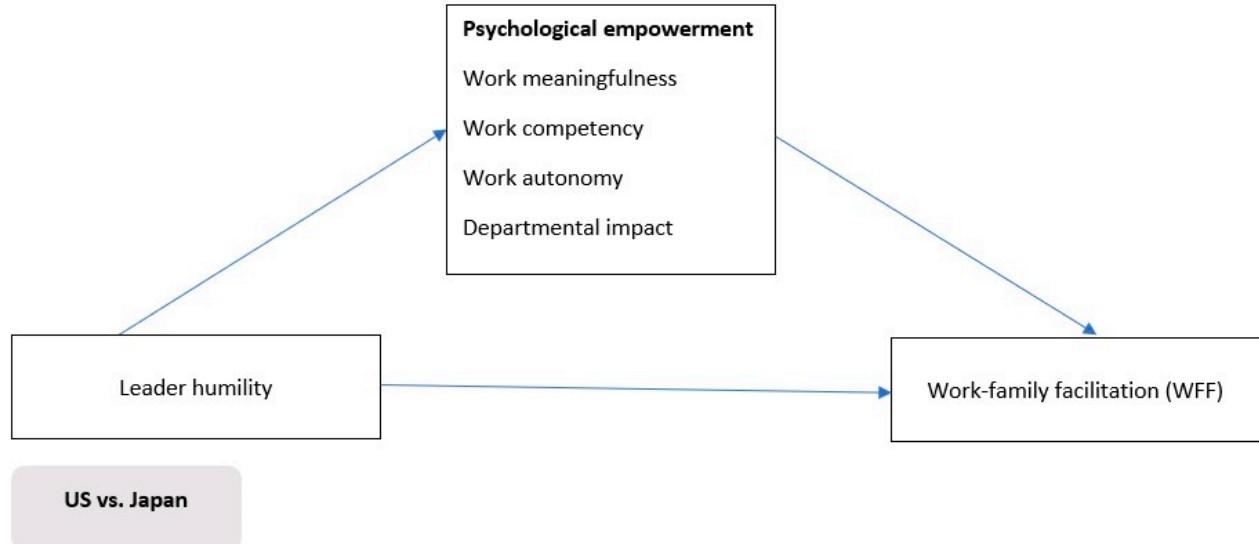

**Figure 1.** Hypothesized research framework.

## 3. Method

### 3.1. Sample and Procedure

All needed Ethics approval was obtained prior to conducting the study. Both the US and the Japanese surveys reminded the potential participants of their rights in survey research and that their participation was voluntary. They were also reminded of their right to withdraw from the study at any point in time without any penalty. The survey data were collected from participants in both the U.S. and Japan using a time-lagged design. There was a gap of two weeks between the two independent administrations of the survey in both contexts. This was conducted to also minimize the causal effect of leader behaviors on employees and the risk of common method bias [52].

The Japanese data were collected by a locally experienced survey agency targeting employees working for large companies accommodating more than 300 employees in the industrial sectors of manufacturing, information, and technology (IT), and wholesale and retail business in the areas of the major big cities of Japan including Tokyo and Osaka.

The first survey included the general demographic background of participants, information about their working experience, and leader-related information including leader humility. Participants who fully completed the first survey were invited to participate in the second survey that included questions about their work attitudes and behaviors. After removing incomplete responses, the final data set included 393 usable data points. Most of the participants (62.3%) were working in the manufacturing industry; 21.7% were from the IT sector, and 16% were from wholesale and retail businesses. The gender of the Japanese data was equally distributed; thus, the number of men and women participants was 192 each. Their average age was 45 years old, and their average working tenure was 22 years. A total of 82.5% of the participants completed their four-year university program and the rest of them completed the post-graduate program. The U.S. data were collected in a similar fashion but using the Prolific platform targeting employees working in the same industrial sectors as the Japanese respondents. All those who completed the first survey were requested to participate in the second one. The final dataset was 132 after deleting the incomplete data points. Most of the U.S. participants were men (66.8%). A total of 46.3% of the participants were in their thirties; 28% were in their 20 s and 25.7% were in

their 40 s. Similar to the Japanese participants, 80% of the U.S. participants completed a four-year university program. The average working tenure was 15.5 years. Missing values were not an issue in the Japanese sample. For the U.S. sample, missing data was related to participants' non-response for the second phase of data collection. Only participants with responses for both Survey 1 and Survey 2 were included in the analyses.

### 3.2. Measures

All main variables were measured using valid and reliable measurement scales, as follows.

*Leader humility* was measured with nine questions developed by [12]. The questions are designed to measure a leader's self-awareness, acknowledge of subordinates' strengths, and display appreciation to subordinates for their sharing knowledge and ideas, and being open to them. Sample questions include "My direct supervisor acknowledges when others have more knowledge and skills than him- or herself", "My direct supervisor shows appreciation for the unique contributions of others", and "My direct supervisor is open to the ideas of others". Strong reliability was exhibited both in the U.S. ($\alpha = 0.93$) and Japan ($\alpha = 0.96$).

*Psychological empowerment:* was measured by the scale developed by Spreitzer (1995). The scale has 12 questions of which three questions each represent a specific subdimension of the concept (i.e., meaning, confidence, autonomy, and impact). The examples for each subdimension are "The work I do is very important to me (meaning)", (alpha U.S. = 0.95 and Japan = 0.95), "I'm self-assured about my capabilities to perform my work activities (confidence)", (alpha U.S. = 0.88 and Japan = 0.87) "I have considerable opportunity for independence and freedom in how I do my job (autonomy)", (alpha U.S. = 0.79 and Japan = 0.90), and "I have a great deal of control over what happened in my department (impact)" (alpha U.S. = 0.93 and Japan = 0.92).

*WFF* was measured by adopting the scale developed by [20]. The scale has questions related to the influences of work on developing skills and knowledge, affective emotions, and family capital that make employees to be better family members. The examples include "My involvement in my work helps me to gain knowledge and this helps me be a better family member (work to family development)", "My involvement in my work puts me in a good mood and this helps me be a better family member (work to family affect)", and "My involvement in my work provides me with a sense of accomplishment and this helps me be a better family member". The scale exhibited good reliabilities in both the U.S. ($\alpha = 0.95$) and Japan ($\alpha = 0.96$).

Control variables that are related to psychological empowerment and WFF were also collected on both the US and Japan surveys. Demographic characteristics of individuals and their working experiences were found to be related to both WFF and psychological empowerment [53,54]. Thus, we included gender, age, education level, and the total time of working in the survey and subsequent analyses.

We explain why these controls are relevant specifically:

(1) Gender plays a crucial role in work–family dynamics because of societal expectations, cultural norms, and gender-specific responsibilities, (2) Age is often related to different life stages and career trajectories can impact an individual's work–family experiences and priorities. For example, younger employees may prioritize career development, while older employees may prioritize family responsibilities, (3) Tenure refers to the length of time an individual has been employed within an organization. It can influence work–family interactions as employees gain experience and familiarity with their work environment, and (4) Education tends to reflect an individual's knowledge, skills, and qualifications. It can influence work–family interactions by shaping career opportunities, income levels, and work demands.

## 4. Analysis

### 4.1. Factor Analyses

Exploratory factor analysis was used to assess how items were loaded on scales to ensure unique variables. We conducted a separate factor analysis for each sample. To assess whether data reductions were valid, we computed Bartlett's test for Sphericity which compares the correlation matrix (a matrix of Pearson correlations) to the identity matrix. In other words, it checks if there is a redundancy between variables that can be summarized with some factors. Thus, we conducted Bartlett's Test of Sphericity to make sure that the correlation matrix of the variables in our dataset diverges significantly from the identity matrix to justify the use of factor analysis. The Kaiser–Meyer–Olkin Measure of Sampling Adequacy is equal or greater than 0.60 for both samples (Japan = 0.94; U.S. = 0.86) indicating that the sample used was adequate. If Bartlett's test of sphericity is significant ($p < 0.05$), as it was in both samples ($p < 0.001$), we met the conditions to proceed with the exploratory factor analysis.

We used principal axis factoring for extraction and Promax with Kaiser Normalization factor rotation. Table 1 shows the resulting factors for both samples including the number of items with coefficient alphas for each factor.

**Table 1.** Scales and coefficient alphas for Japanese and American samples.

|  | Japan (*N* = 392) | | United States (*N* = 132) | |
| --- | --- | --- | --- | --- |
|  | **# Items** | **Reliability** | **# Items** | **Reliability** |
| Leader Humility | 12 | 0.96 | 12 | 0.93 |
| Work Meaningfulness | 3 | 0.95 | 3 | 0.95 |
| Work Competence | 3 | 0.88 | 3 | 0.87 |
| Autonomy | 3 | 0.90 | 3 | 0.79 |
| Departmental Impact | 3 | 0.92 | 3 | 0.93 |
| Work–family Facilitation | 9 | 0.96 | 6 | 0.95 |

### 4.2. Common Method Bias

We collected our data at two measurement points to minimize Common Method Bias [55]. During time 1, participants were asked to report their perceptions of leader humility, their work meaningfulness, work competence, work autonomy, and their departmental impact. During time 2, which was approximately two weeks after time 1, the participants were asked to report their levels of WFF. Each participant was assigned an identifying number during time 1 which enabled us to match the participants during time 2 data collection. Additionally, the factor analyses revealed that the first factor extracted accounted for less than 50% of the total variance for both samples (Japan = 36.03% and U.S. = 28.90%) suggesting a lower likelihood of common method bias.

We utilized SPSS V. 28 as the main statistical package and was used to test the direct effects of leader humility using correlation and regression functions with control variables. We employed Hayes' PROCESS Macro V. 4.1 [56] to assess the mediation effects in our study. This macro is a routine in the SPPS regression function and is specifically designed for conducting mediation analyses, which examine the underlying mechanisms through which an independent variable affects a dependent variable. Model 4 is specifically used to estimate mediation effects. The combination of SPSS and Hayes' PROCESS Macro V. 4.1 enabled us to conduct robust statistical analyses and gain meaningful insights into the mediation effects of leader humility.

## 5. Results

Tables 2 and 3 report the reliabilities and intercorrelations of the study variables for both samples. Most of the variables were significantly correlated and in the expected direction.

**Table 2.** Means, standard deviations, and Pearson correlations for the U.S. Sample.

| | 1 | 2 | 3 | 4 | 5 | 6 | 7 | 8 | 9 | 10 |
|---|---|---|---|---|---|---|---|---|---|---|
| 1. Gender | -- | | | | | | | | | |
| 2. Age | −0.05 ** | -- | | | | | | | | |
| 3. Tenure | −0.09 * | 0.83 ** | -- | | | | | | | |
| 4. Education | 0.08 | 0.16 | 0.06 | -- | | | | | | |
| 5. Leader Humility | 0.00 | 0.09 | 0.05 | 0.26 ** | -- | | | | | |
| 6. Work Meaningfulness | −0.06 | 0.21 * | 0.18 * | 0.07 | 0.38 ** | -- | | | | |
| 7. Work Competence | 0.08 | 0.23 ** | 0.17 | 0.04 | 0.15 | 0.35 ** | -- | | | |
| 8. Work Autonomy | −0.06 | 0.09 | 0.02 | 0.13 | 0.38 ** | 0.36 ** | 0.44 ** | -- | | |
| 9. Departmental Impact | −0.16 | 0.31 ** | 0.34 ** | 0.14 | 0.33 ** | 0.50 ** | 0.35 ** | 0.36 ** | -- | |
| 10. WFF | −0.09 | 0.28 ** | 0.19 * | 0.17 * | 0.29 ** | 0.60 ** | 0.38 ** | 0.18 * | 0.37 ** | |
| Mean | 1.33 | 1.98. | 15.64 | 1.20 | 4.25 | 3.62 | 4.45 | 4.08 | 3.66 | 3.42 |
| SD | 0.47 | 0.74 | 8.00 | 0.40 | 0.53 | 1.27 | 0.73 | 0.82 | 0.77 | 0.98 |

* $p < 0.05$. ** $p < 0.01$ (2-tailed). $N = 132$

Note. WFF = work–family facilitation. $N$ for WFF was 206.

**Table 3.** Means, standard deviations, and Pearson correlations for the Japanese sample.

| | 1 | 2 | 3 | 4 | 5 | 6 | 7 | 8 | 9 | 10 |
|---|---|---|---|---|---|---|---|---|---|---|
| 1. Gender | -- | | | | | | | | | |
| 2. Age | −0.48 ** | -- | | | | | | | | |
| 3. Tenure | −0.40 ** | 0.72 ** | -- | | | | | | | |
| 4. Education | −0.05 | 0.00 | −0.06 | -- | | | | | | |
| 5. Leader Humility | 0.01 | −0.03 | 0.07 | 0.09 | -- | | | | | |
| 6. Work Meaningfulness | −0.03 | 0.02 | 0.03 | −0.01 | 0.42 ** | -- | | | | |
| 7. Work Competence | −0.03 | 0.23 ** | 0.04 | 0.06 | 0.32 ** | 0.54 ** | -- | | | |
| 8. Work Autonomy | −0.02 | 0.10 * | 0.09 | 0.13 * | 0.46 ** | 0.51 ** | 0.48 ** | -- | | |
| 9. Departmental Impact | −0.16 ** | 0.09 | 0.08 | 0.08 | 0.41 ** | 0.56 ** | 0.52 ** | 0.58 ** | -- | |
| 10. WFF | −0.03 | 0.12 * | 0.05 | 0.08 | 0.27 ** | 0.35 ** | 0.30 ** | 0.28 ** | 0.31 ** | |
| Mean | 1.50 | 45.64 | 15.64 | 1.20 | 4.25 | 3.62 | 4.45 | 4.08 | 3.66 | 3.42 |
| SD | 0.50 | 9.56 | 8.00 | 0.40 | 0.53 | 1.27 | 0.73 | 0.82 | 0.77 | 0.98 |

* $p < 0.05$. ** $p < 0.01$ (2-tailed). $N = 392$

Note. WFF = work–family facilitation.

### 5.1. Direct Effects of Leader Humility

Hypothesis 1 predicted the leader humility would be positively related to WFF in Japan and the United States. First, we computed Pearson correlations for each sample: $r_J = 0.27$ ($p < 0.001$) and $r_{us} = 0.27$ ($p < 0.001$). Next, conducted a more robust test using linear regression to by entering the control variables (i.e., gender, age, tenure, and education) in the equation. The controls were entered first followed by leader humility. In each case, the controls entered as a block were significant, $F_J(4, 387) = 2.27$, $p < 0.05$), an $F_{US}(4, 129) = 3.67$, $p < 0.01$). Similarly, the inclusion of leader humility was significant over and above the control variables: $F_J(5, 386) = 8.22$, $p < 0.001$), an $F_{US}(5, 128) = 5.04$, $p < 0.001$). Finally, the regression coefficients with controls were also significant for both samples ($\beta_J = 0.26$, $p < 0.001$) and ($\beta_{US} = 0.26$, $p < 0.002$). Thus, Hypothesis 1 was supported.

### 5.2. Mediating Role of Leader Humility

Mediation analyses were conducted using Hayes' PROCESS Macro V. 4.1 (Hayes, 2018) Model 4. Each analysis used 10,000 bootstrap samples and a 95% confidence interval. Completely standardized indirect effects are reported for each hypothesis. As noted above, all analyses controlled for gender, age, tenure, and education.

Recall that H2 predicted that meaningful work mediated the relationship between leader humility and WFF in the U.S. and Japan. The direct effect of leader humility on

WFF was $\beta_J = 0.27$, and $p < 0.001$ was significant for the Japanese sample but not the U.S. sample ($\beta_{US} = 0.05$, $p > 0.05$). Based on the recommended bias-corrected bootstrapping by Hayes (2018), meaningful work significantly mediated the relationship between humility and WFF for both samples: indirect effects for Japan (indirect effect$_J$ = 0.12, $p < 0.01$, 95% CI = [0.07, 0.18]) and the U.S. (indirect effect$_{US}$ = 0.19, $p < 0.01$, 95% CI = [0.09, 0.31]). Overall, strong support was found for H2 that meaningful work is a significant mediator of leader humility—WFF relationship for both samples.

Next, the Hypothesis (H3) that the relationship between leader humility and WFF through work competence was tested. Leader humility positively related to WFF for the Japanese ($\beta_J = 0.19$, $p < 0.001$) and U.S. samples ($\beta_{US} = 0.20$, $p < 0.05$). The indirect effect of competence was significant for Japan (indirect effect$_J$ = 0.07, $p < 0.01$, 95% CI = [0.03, 0.12]) but not for the U.S. (indirect effect$_{US}$ = 0.03, $p > 0.05$, 95% CI = [−0.01, 0.07]) because the confidence interval included zero. Thus, H3 was supported for the Japanese but not the U.S. sample.

Similarly, we tested whether autonomy mediated the relationship between leader humility and WFF in the U.S. and Japan (H4). Leader humility positively related to WFF for the Japanese ($\beta_J = 0.17$, $p < 0.001$) and U.S. samples ($\beta_{US} = 0.20$, $p < 0.05$). Note once again that the indirect effect of autonomy was significant for Japan (indirect effect$_J$ = 0.08, $p < 0.01$, 95% CI = [0.02, 0.15]) but not for the U.S. (indirect effect$_{US}$ = 0.03, $p > 0.05$, 95% CI = [−0.08, 0.11]) because the confidence interval included zero. Thus, H4 was supported for the Japanese but not the U.S. sample.

We then examined whether departmental impact mediated the relationship between leader humility affecting WFF in the U.S. and Japan (H5). Leader humility was positively related to WFF for the Japanese ($\beta_J = 0.17$, $p < 0.001$) and U.S. samples ($\beta_{US} = 0.23$, $p < 0.05$). Note that the indirect effect of departmental impact was significant for Japan (indirect effect$_J$ = 0.08, $p < 0.01$, 95% CI = [0.04, 0.15]) and for the U.S. (indirect effect$_{US}$ = 0.06, $p < 0.05$, 95% CI = [0.01, 0.14]). Thus, H5 was supported both for the Japanese and the U.S. sample.

Finally, in addition to the tests of the hypotheses, we performed the most robust test of mediation between leader humility and WFF. Since the mediator variables were correlated, we wanted to ascertain whether mediation would occur when we controlled for the effects of the other mediators. In this case, in addition to the four control variables, we entered all mediator variables as covariates. The results indicated that 'work meaningfulness' was the only significant mediator when controlling for all others, a result that held for both samples: for Japan (indirect effect$_J$ = 0.08, $p < 0.01$, 95% CI = [0.02, 0.15]) and for the U.S. (indirect effect$_{US}$ = 0.17, $p < 0.05$, 95% CI = [0.08, 0.31]). Finally, since gender is an important variable in WFF, the effects of gender on the findings were examined. None of the analyses were statistically significant.

## 6. Discussion

The findings offer robust support for the notion that leader humility fosters positive WFF in both the United States and Japan, thereby affirming Hypothesis 1. The correlation and linear regression analyses collectively highlight the significance of leader humility in fostering WFF. Research has found that family-supportive supervisors tend to play an important role in facilitating WFF (for example, [57]). In a similar vein, leader humility could potentially create an environment where employees feel more comfortable in approaching their leader about family-related matters, regardless of cultural context. Employees might feel that their leader would be understanding of their family obligations. In fact, leader support can mitigate employees' stress levels and work–family obligations [58]. Thus, the support for this hypothesis makes sense and this relationship underscores the potential of leader humility as an effective lever to foster WFF in varying cultural contexts, which, in turn, may contribute to positive outcomes such as overall job satisfaction and performance of employees. Research does indicate the positive influence of leader humility on employee wellbeing [8].

Our findings lend support to the notion that meaningful work serves as a mediator of the relationship between leader humility and WFF, as posited in Hypothesis 2. This indicates that perceiving work as meaningful may play an essential role in how leader humility affects WFF across diverse cultural contexts. Work meaning was also a substantial mediator between leader humility and WFF when controlling for other mediators in the sample. Work meaningfulness thus may be a particularly successful avenue for leaders to improve employees' WFF. This implies the likelihood that finding meaning in one's work or finding that one's job tasks make a significant contribution can put the individuals in a positive mood which aids also in WFF. Several research studies indicate the important role of meaningful work in positive outcomes for employees (e.g., [59]).

Contrarily, the indirect effects of work competency and autonomy were found to significantly mediate the relationship between leader humility and WFF in the Japanese context not in the U.S. (Hypotheses 3 and 4, respectively). This finding may point to the potential cultural nuances that shape how leader humility affects WFF, warranting further cross-cultural investigation. It is plausible that these two work dimensions are inherent parts of the jobs we conducted this study on among the US sample and, therefore, leader humility may have less of a significant impact. For instance, autonomy and a sense of competence may not necessarily be tied to the leader's disposition if the job incumbents have these features as part of their jobs. Further, the effectiveness of a positive, collaborative environment on WFF across different cultures suggests that certain psychological needs or aspects of human social cognition are universal. According to the Self-Determination Theory [60], all humans, regardless of cultural backgrounds, have innate needs for competence, autonomy, and relatedness. A collaborative environment may satisfy these needs by empowering individuals, increasing their autonomy, and fostering connections with colleagues. Thus, though leader humility may itself affect WFF among the U.S. sample, certain job features or employees' personal traits may be more influential to augment autonomy and competence rather than such leader traits in the U.S. Future research should be warranted to explore these ideas more fully by investigating the potential job- and individual-related factors that influence such components of psychological empowerment.

On a similar note, the mediation analysis for departmental impact showed that it significantly mediated the relationship between leader humility and WFF in both Japan and the U.S., offering support for Hypothesis 5. This result indicates that the impact of humble leaders can permeate organizational boundaries, affecting employees' experiences in their work and family domains. Humble leaders can, in keeping with SIP theoretical tenets, bring about a collaborative and supportive climate for their employees in which employees' inputs are valued, augmenting their feeling of being embraced and impactful in their working environment. We found that a positive, collaborative environment that transcends cultural differences can boost WFF, thus underlining the possible pragmatic and productive outcomes. Broadly speaking, these insights increase our comprehension of the way that leader humility can beneficially affect employees' professional and personal lives. The findings afford possibilities for subsequent research as well as offering practical considerations for nurturing leadership and promoting employee wellness.

## 7. Theoretical and Practical Implications

Our study provides significant theoretical contributions. First, the study adds to our understanding of the tenets of SIP theory which refers to the important role that a leader performs by giving cues about the organization to their subordinates. It further substantiates the pivotal role a leader can play in bringing about outcomes that meet organizational and employee needs, such as WFF. Our study also corroborates that a leader's personal dispositions, such as humility, can have an impact on his or her employees' psychological status, shaping positive attitudes toward their work. Employees' psychological factors have significant long-term effects on organizations by enhancing their loyalty and commitment, thus resulting in sustainable employability that benefits organizations [61,62]. Moreover, the positive link between leader humility and WFF implies that leader humility

enhances employees' wellbeing in life by enhancing their confidence and satisfaction in their family life. The finding confirms the significant managerial effect of leader humility, supporting the findings of previous studies [4,10,28]. Furthermore, the findings clarify the crucial benefits of leader humility to both work and life domains. Thus, our study has theoretical contributions as well, underlining the important factor of leadership traits in forming employees' positive psychological work and life attitudes that are essential for organizational sustainability.

Second, this study sheds new light on our understanding of the concept of leader humility by examining its effects on WFF in two culturally distinct contexts, the United States and Japan. Although the humility concept is accepted mostly in the Asian context [3,4,10], the positive effect of leader humility on employees in the U.S. confirms the general acceptance and effectiveness of the concept, regardless of cultural contexts. This implies that a leader's desired prototype may be changing in the U.S. in accordance with the changing demographic shifts and newly emerging values among employees. These values suggest embracing inclusive attitudes from leaders rather than the deterministic or masculine model typically associated with leaders in the West, thereby confirming the effectiveness of humility on WFF in both East and West. As such, this cross-cultural exploration adds a valuable dimension to the existing research on leader humility and its influence on various work environments, suggesting the efficacy of leader humility regardless of cultural differences. Furthermore, the findings offer unique insights into the mediating roles that meaningful work and departmental impact can play in linking leader humility to WFF in both contexts. The identification of these potential mediators corroborates the prior work [40] and expands the contextual boundaries to the West by highlighting its practical applicability regardless of cultural contexts.

On a practical level, this study underscores the importance of nurturing humility in leaders. Organizations may benefit from integrating humility training into their leadership development programs, which can lead to an enrichment in WFF among their employees. Such practices are critical in sustaining organizations. By establishing a constructive relationship between leader humility and WFF, the study suggests that humble leadership could be instrumental in promoting healthier work–life interactions among employees, ultimately leading to higher job satisfaction, job performance, and overall employee wellbeing. The variances in the mediating factors between the U.S. and Japan emphasize that organizations operating in different cultural contexts may need to adapt their strategies accordingly. For instance, in a Japanese work context, emphasizing the promotion of autonomy and departmental impact may yield more substantial results. Finally, the pivotal mediating role of meaningful work indicates that fostering a sense of purpose and meaning in work can significantly improve WFF, which can, in turn, elevate employee engagement and productivity.

## 8. Limitation and Future Research

There are several possible limitations to this study. First, the cross-sectional nature of this study limits the inference of causal relationships among the variables. Even though we found significant associations between leader humility and WFF, these results should be interpreted with caution. Future studies might employ longitudinal or experimental designs to establish a clear causal relationship between these variables. For example, there could be a two-wave study where phase one would collect data on employee perception of their leader's humility, and phase two would include data collection on how such humility would have an effect on work–family facilitation of the employees over a longer period of time. Further, our study largely depends on self-reports that might contribute to response bias and overstate the relationships among variables. The disparity in sample size between the two countries could raise questions about sampling and measurement equity. Notwithstanding the consistent data collection procedure, the U.S. study achieved a notably lower participation rate in the second survey. Thus, the degree of similarity of the samples cannot be definitively ascertained. Although the samples from the U.S. and Japan

might offer valuable insights from a comparative standpoint, such samples are unlikely to fully capture the cultural diversity being studied. Finally, questionnaires may not fully measure the influence of leader humility on WFF across different cultures like the U.S. and Japan.

Second, the study relies on self-reported measures, which could potentially introduce response bias and inflate the relationships among variables. In addition, the difference in the number of samples between the two countries may occur as an issue related to the equity in sampling and measurement. Although the same approach was applied in collecting data, the participation rate to the second survey was far low in the U.S. Therefore, future research could benefit from incorporating multiple sources of data, such as peer or supervisor evaluations, ensuring the equity in the number of samples to validate the findings. Third, while our samples from the U.S. and Japan provide useful comparative perspectives, they cannot represent the full diversity of cultures worldwide. Additional research across different cultural contexts would enrich our understanding of how leader humility impacts WFF in various cultural settings. Lastly, our study has focused on only a few potential mediators (meaningful work, autonomy, and departmental impact) in the relationship between leader humility and WFF. There may be other relevant factors such as team cohesion [1], organizational culture, or individual traits that could further mediate this relationship. Future research should continue to explore these and other potential mediators.

There are several directions for future research. First, researchers could extend this work by examining how different leadership styles interact with leader humility to influence WFF. For example, transactional or task-oriented leadership may have a distinct impact on leader humility which could in turn affect its influence on WFF. Different leadership styles may impact the dimensions of psychological empowerment in their own unique manner bringing it a set of results distinct from what we observed in the current study. Second, we also persuade researchers to incorporate additional variables in the study to further our understanding of the impact of leader humility. For example, exploring how individual characteristics (e.g., personality traits and personal values) moderate the effects of leader humility on WFF could also provide a more nuanced understanding of this relationship.

Third, different aspects of the leader's personality or their unique style of delegation may also impact how their subordinates perceive the leader's humble actions. Future research should ideally explore such interrelated dynamics to fully uncover the concept of leader humility. Fourth, as the world is increasingly becoming smaller and more diverse, a common feature of many workplaces, it would be interesting to study how people with differing ethnic backgrounds and working in a Western or Eastern culture may appraise leader humility. For instance, do people from Asian cultures working in the U.S. or people from Western cultures working in Asian countries regard leader humility differently? Finally, researchers could also investigate the potential moderating role of organizational culture or the type of jobs held to determine whether certain cultural values or job responsibilities may enhance or mitigate the effects of leader humility on WFF.

## 9. Conclusions

This study contributes to our insight of the way that leader humility affects WFF in two diverse cultural context—the United States and Japan. Leader humility was found to be positively related to WFF in both countries, mediated by the components of psychological empowerment. Notably, meaningful work emerged as the most significant mediator that links the relationship leader humility and WFF. The findings underscore the potential of leader humility in enhancing employees' work and family lives, by fostering positive work–family interactions and contributing to organizational sustainability. Although our study provides new insights into the role of leader humility, it underscores the complexity of this relationship, given the variety of mediating factors and their cultural manifestations. Forthcoming research should continue to investigate the intricate role of leaders' humility to gain a better estimation of how and under what conditions a leader's humility can

enhance WFF. Overall, our study provides promising directions for further research on the dynamics of leadership and WFF in diverse cultural settings.

**Author Contributions:** Conceptualization, N.G. and S.K.; methodology, N.B.; software, N.B.; validation, N.B., S.K. and N.G.; formal analysis, N.B.; investigation, S.K. and N.G.; resources, N.B., S.K. and N.G.; data curation, N.G., S.K. and N.G.; writing—original draft preparation, S.K., N.G. and N.B.; writing—review and editing, S.K., N.G. and N.B.; visualization, S.K., N.G. and N.B.; supervision, S.K. and N.G.; project administration, S.K. and N.G.; funding acquisition, S.K. All authors have read and agreed to the published version of the manuscript.

**Funding:** This research was partially funded by JSPS KAKENHI, grant number 22K01732.

**Institutional Review Board Statement:** The study was approved by the Institutional Review Board (or Ethics Committee) of the University of Redlands (IRB approval 2021-01-Redlands).

**Informed Consent Statement:** Informed consent was obtained from all subjects involved in the study.

**Data Availability Statement:** Data is available from the authors upon reasonable requests.

**Conflicts of Interest:** The authors declare no conflict of interest.

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
