# Peer review of "Sustainability through Humility: The Impact of Humble Leadership on Work–Family Facilitation in the U.S. and Japan"

_sustainability, doi:10.3390/su151914367_

Round 1

Reviewer 1 Report

Dear Authors,

The study is a novel addition to the body of knowledge. It is well-written and well-explained. There are certain points you need to address: 

1. Please shorten the title if possible. 

2. Introduction is well-written. However, you need to organize a little more. The literature review is portraying sound theoretical background. However, you have not discussed confounding variables (e.g., gender) in the introduction and literature review. 

3. Please elaborate on sample characteristics. The audience want to know who were the participants. Explain the demographic characteristics numerically.  

4. In table 1, you have written it is the result of EFA, but it is just reliability presented in the table. Please change the description of table 1 to reliability results. 

5. In the results, please provide the results of the control (gender) variable. 

6. Discussion needs extensive revision. Please explain your results with reference to previous research and also in the context of US and Japan. You have not provided a single related research for your findings. You have referred to related research with theoretical and practical implications but you need to give reference in discussion. Please read discussion part of the articles related to your research and take the guidance on how to discuss results in detail. 

All other is very good.

Good Luck! 

Please apply Grammarly premium version to your article. It is enough for English editing. 

Author Response

Dear reviewer,

Thank you so much for your valuable feedback. We are truly grateful for your comments as they have helped us to improve the paper. Below, we have listed each of your comments and provided our response. We have included the new information on the manuscript, wherever needed, in red font so that they are easily identifiable for you. Once again, thank you.

  1. Please shorten the title if possible.

RESPONSE: We have done this. 

2. Introduction is well-written. However, you need to organize a little more. The literature review is portraying sound theoretical background. However, you have not discussed confounding variables (e.g., gender) in the introduction and literature review. 

RESPONSE:  Thank you for this comment. We have included information on ‘gender’ and other control variables in the ‘introduction’ and ‘literature review’ sections

3. Please elaborate on sample characteristics. The audience want to know who were the participants. Explain the demographic characteristics numerically.  

RESPONSE: The demographic characteristics of both samples regarding their gender, age, tenure, and educational background were added.

4. In table 1, you have written it is the result of EFA, but it is just reliability presented in the table. Please change the description of table 1 to reliability results. 

RESPONSE: We have changed the description to reliability results as suggested. 

5. In the results, please provide the results of the control (gender) variable. 

RESPONSE: We controlled for gender (and other controls) in our analyses. Gender did not have a statistically significant impact on any of the results in for the Japanese or the U.S. samples. 

6. Discussion needs extensive revision. Please explain your results with reference to previous research and also in the context of US and Japan. You have not provided a single related research for your findings. You have referred to related research with theoretical and practical implications but you need to give reference in discussion. Please read discussion part of the articles related to your research and take the guidance on how to discuss results in detail. 

RESPONSE:  Thank you for this comment. We have included new references in the ‘discussion’ section.

All other is very good.

Reviewer 2 Report

I have carefully reviewed the manuscript, titled “Sustainability through Humility: How Humble Leadership Affects Work-Family Facilitation via Psychological Empowerment in U.S. and Japan”. The study was to examine the influence of leader humility on work-family facilitation (WFF) in the U.S. and Japan by exploring the mediating roles of the four dimensions of psychological empowerment (meaningful work, autonomy, competency, and impact) on this relationship.

The study has been substantially revised and some strong points (clear introduction, accurate statistical analysis, constructive discussion).

However, I would like to ask the authors to address some points in order to improve the paper:

Introduction:

1) Can you present some psychological explanations for meaning at work? Why do people experience it in general? What are its underlying mechanisms (p. 5)

2) As regards H2, what does meaningful work comprise>

Method:

3) Please, provide Cronbach’s coefficients for all your measures.

4) How did you handle missing values in your data? (If any exist)

Results:

5) The results are properly showed.

Discussion:

6) What are the underlying mechanisms responsible for this result: “Thus, though leader humility may itself affect WFF among the U.S. sample, certain job features or employees’ personal traits may be more influential to augment autonomy and competence rather than such leader trait in the U.S.” (p. 12)?

7) Can you elaborate on the following statement: “We found that a positive, collaborative environment that transcends cultural differences can boost WFF thus underlining the possible pragmatic and productive outcomes.”. Please, provide a potential explanation of this statement as it is an interesting result. Is it a matter of cognitions  or something else?

8) What kind of future research with longitudinal design would you propose? (Limitations).

Author Response

Dear reviewer,

Thank you so much for your valuable feedback. We are truly grateful for your comments as they have helped us to improve the paper. Below, we have listed each of your comments and provided our response. We have included the new information on the manuscript, wherever needed, in red font so that they are easily identifiable for you. Once again, thank you.

1) Can you present some psychological explanations for meaning at work? Why do people experience it in general? What are its underlying mechanisms (p. 5) 

RESPONSE: Psychologically, meaningful work allows individuals to express their values, beliefs, and interests, thus contributing to a sense of authenticity and purpose. Having control over one's work and the freedom to make decisions adds to this sense, by fostering ownership and responsibility. Engaging in complex tasks that require creative problem-solving and skill application can lead to achievement and fulfillment, nurturing personal growth and mastery. We have now mentioned all these in the section devoted to ‘psychological empowerment.’ 

2) As regards H2, what does meaningful work comprise 

 RESPONSE: As defined by Spreitze (1995), meaningful work comprises the value of work purpose and goal. Work meaningfulness is increased when employees’ work goals are congruent with their value system and expected ideas about the work. To enhance the clarification of each composition of psychological empowerment including meaningful work, we added these explanations about each dimension of psychological empowerment in the section on ‘psychological empowerment.’  

 3) Please, provide Cronbach’s coefficients for all your measures.  

RESPONSE: We have added Cronbach alphas for each of our measures in the Method section. 

4) How did you handle missing values in your data? (If any exist)   

RESPONSE: Missing values were not an issue in the Japanese sample. For the U.S. sample missing data was related to participants non-response for the second phase of data collection. Only participants with responses for both Time 1 and Time 2 were included in the analyses. We have mentioned this now in Methods section. 

 5) What are the underlying mechanisms responsible for this result: “Thus, though leader humility may itself affect WFF among the U.S. sample, certain job features or employees’ personal traits may be more influential to augment autonomy and competence rather than such leader trait in the U.S.” (p. 12)?   

 RESPONSE: We explain this finding based on Ryan and Deci’s (2000) Self-Determination Theory. The effectiveness of a positive, collaborative environment on WFF across different cultures suggests that certain psychological needs or aspects of human social cognition are universal. According to the Self-Determination Theory, all humans, regardless of cultural backgrounds, have innate needs for competence, autonomy, and relatedness. A collaborative environment may satisfy these needs by empowering individuals, granting them autonomy, and fostering connections with colleagues. 

 6) Can you elaborate on the following statement: “We found that a positive, collaborative environment that transcends cultural differences can boost WFF thus underlining the possible pragmatic and productive outcomes.” Please, provide a potential explanation of this statement as it is an interesting result. Is it a matter of cognitions or something else?  

RESPONSE: Thank you for the encouraging comment. We have edited the beginning of the sentence to indicate that we were reporting what the study tended to indicate.  

7) What kind of future research with longitudinal design would you propose? (Limitations). 

 RESPONSE: Thank you for this suggestion. We have provided an example of a longitudinal study in the section on ‘future directions.’   

Reviewer 3 Report

This is a well written paper on an interesting topic, and your writing style is clear, fluent and easy to read. 

The paper does, however, have a significant issue in that the generalisations about the US and Japan are based on a very small sample size and, in my view, the claims made about the data providing evidence for the congruence between Japan and the US cannot currently be substantiated.

The intorduction section is clear and the literature review well defined, although I would suggest a little more critical analysis to more clearly highlight the research gaps that this paper will start to fill.

Unfortunately, there is insufficient information in the methods section to provide the reader with any confidence that the organisations/businesses sampled were comparable in any way in terms of sector, size, industry or geography. Neither is there any information on how the employees compared in terms of key demographic variables including gender, family context, length of employment, etc. Response rates (%) for the surveys are also not reported. You do reflect on these issues, in a well written papragraph (section 8) but not until after the end of the discussion section.

Having said that, the paper could be very successfully edited by adding in much more information regarding the methods, and including a clear statement in the introduction, acknowledging the limitations of the study (as per section 8), and casting this paper as a first step in researching this complex topic.

This paper is very well written in terms of English language, and I have only made a couple of minor suggestions for typo and clarity of expression edits. The authors have a nice clear and flowing style. 

Author Response

Dear reviewer,

Thank you so much for your valuable feedback. We are truly grateful for your comments as they have helped us to improve the paper. Below, we have listed each of your comments and provided our response. We have included the new information on the manuscript, wherever needed, in red font so that they are easily identifiable for you. Once again, thank you.

  1. The paper does, however, have a significant issue in that the generalisations about the US and Japan are based on a very small sample size and, in my view, the claims made about the data providing evidence for the congruence between Japan and the US cannot currently be substantiated. 

RESPONSE: We acknowledge that the sample size is not equal for the US and Japanese data. Hence we do address this as a limitation and advice caution of generalizing results. However, we did collect data on same industries and had the same screening questions which would provide more comparative data.  

2.The introduction section is clear and the literature review well defined, although I would suggest a little more critical analysis to more clearly highlight the research gaps that this paper will start to fill.  

RESPONSE: We appreciate this comment and have now included a more critical review of the research gaps and how our research  

 3. Unfortunately, there is insufficient information in the methods section to provide the reader with any confidence that the organisations/businesses sampled were comparable in any way in terms of sector, size, industry or geography. Neither is there any information on how the employees compared in terms of key demographic variables including gender, family context, length of employment, etc. Response rates (%) for the surveys are also not reported. You do reflect on these issues, in a well-written papragraph (section 8) but not until after the end of the discussion section. 

 RESPONSE: We apologize for this overlook. Our data were collected from USA and from Japan. We have included information on demographic variables, wherever they were available from our survey.  

4. Having said that, the paper could be very successfully edited by adding in much more information regarding the methods, and including a clear statement in the introduction, acknowledging the limitations of the study (as per section 8), and casting this paper as a first step in researching this complex topic.

 RESPONSE: Thank you for this suggestion. We have included the statement about the purpose of this study as the beginning sentences in this manuscript. We have also provided information asked in the ‘methods’ section, acknowledged the ‘limitations’ in this study and included a statement that our study is innovative in the ‘introduction’ section (in the paragraph on the ‘three goals’ of this study).  

Round 2

Reviewer 3 Report

The authors have made the changes suggested and the paper is much clearer and the arguments more convincing as a result. There are just a few minor suggestions for amendments to polish the writing further. 

Nice clear and well written text. I've just made a few grammatical suggestions in the attached file. 

Author Response

Dear Reviewer, 

Thank you for your valuable input into improving the English language of our paper. We inserted all your suggestions using red font but in yellow shade, so that the R and R2 edits are easily identifiable for you. 

Once again, thank you from all of us in the team who wrote this paper. 

Sincerely,

The authors. 
